# No mass extinction for land plants at the Permian–Triassic transition

Hendrik Nowak [1], Elke Schneebeli-Hermann [2] & Evelyn Kustatscher[1,3]

The most severe mass extinction among animals took place in the latest Permian (ca. 252 million years ago). Due to scarce and impoverished fossil floras from the earliest Triassic, the common perception has been that land plants likewise suffered a mass extinction, but doubts remained. Here we use global occurrence data of both plant macro- and microfossils to analyse plant biodiversity development across the Permian–Triassic boundary. We show that the plant fossil record is strongly biased and that evidence for a mass extinction among plants in the latest Permian is not robust. The taxonomic diversities of gymnosperm macrofossils and of the pollen produced by this group are particularly incongruent. Our results indicate that gymnosperm macrofossils are considerably undersampled for the Early Triassic, which creates the impression of increased gymnosperm extinction in the latest Permian.

[1] Museum of Nature South Tyrol, Bindergasse/Via Bottai 1, Bozen, Bolzano 39100, Italy. [2] Paläontologisches Institut und Museum, Karl Schmid-Strasse 4, 8006 Zürich, Switzerland. [3] Department für Geo- und Umweltwissenschaften, Paläontologie und Geobiologie, Ludwig–Maximilians–Universität, and Bayerische Staatssammlung für Paläontologie und Geologie, Richard–Wagner–Straße 10, 80333 Munich, Germany. Correspondence and requests for materials should be addressed to H.N. (email: hendrik.nowak@naturmuseum.it)

The end-Permian mass extinction was the most severe extinction event in the Phanerozoic, with an estimated loss of ca. 80–96% of species and ca. 50% of families of marine invertebrates[1,2]. On land, tetrapods[3] and insects[4] were likewise diminished and also for plants a loss of diversity (or taxonomic richness) has been suggested to occur between the Changhsingian (latest Permian) and the Induan (earliest Triassic), with a magnitude that is comparable to the losses in marine invertebrates[5–7]. However, it has been questioned whether the terrestrial and marine events are coeval[8–11]. The fossil floras from the Early Triassic are marked by impoverishment and on the northern hemisphere often by the dominance of lycopsids (club mosses), especially *Pleuromeia*[12–14]. Correspondingly, spores of lycopsids and ferns[13] often dominate sporomorph (spores and pollen) assemblages from this time. This is interpreted as a period of survival, which is followed by the recovery of conifers starting in the Olenekian (late Early Triassic) at the Smithian-Spathian boundary[13,15,16]. In the southern hemisphere (Gondwana), glossopterids were dominant in the Permian, but they were diminished at the Permian–Triassic boundary and replaced by the *Dicroidium* flora[17,18]. Lower Triassic successions yielded only a reduced number of well-preserved macrofossils of land plants and are also notably void of coal measures[19]. The dearth of fossils agrees seemingly with a massive loss of vegetation, but it could also be accounted for by a severe taphonomic bias.

Local and regional studies have repeatedly delivered results that raise doubts about the importance of the end-Permian event for land plants[20–25]. Data on sporomorphs from the Permian–Triassic boundary interval are much more abundant than macrofossils, providing valuable information about the development of terrestrial floras during this critical time. The potential of these datasets is still mostly unused, for in most cases, sporomorph diversity has only been studied on local or regional scales until now.

In order to gain a more coherent picture of land plant history, a detailed, comparative assessment of the stratigraphic ranges and diversities of sporomorph and macrofossil taxa from the Lopingian (upper Permian) to the Middle Triassic is presented here. We show that the extinction of land plants at the Permian–Triassic boundary was much less severe than previously thought and that the apparent mass extinction can essentially be explained by the dearth of data from the Lower Triassic. The fossil records of sporomorphs and macrofossils show considerable differences, primarily concerning gymnosperms, which might in part be attributed to taphonomic bias.

## Results

**Macrofossil record.** Plant macrofossil species (Supplementary Data 1) have their highest diversity at the base of the studied stratigraphic interval, in the Wuchiapingian (Fig. 1, Supplementary Data 2). Their diversity declines towards the Changhsingian, followed by a loss of more than half of the species across the Permian–Triassic boundary. Species diversity starts to recover in the Olenekian and continues towards the Ladinian. This pattern would seem to conform to previous results based on plant macrofossil genera and families and common expectations (see refs. [5–7]). However, on the genus level, diversity loss across the Permian–Triassic boundary is not catastrophic and diversity even increases slightly between the Wuchiapingian and the Changhsingian. In general, the genus diversity curve is rather flat. Even more important, genus originations in the Changhsingian exceed extinctions, which means that net diversity increases during this stage. Extinctions only exceed originations in the Induan, which is also the stage with the highest number of extinction events (Fig. 2). Conversely, origination numbers are very low, which results in the overall lower total diversity in the Induan. The normalized diversity curve suggests that the mean standing diversity declined far less (Fig. 3). The results of shareholder quorum subsampling indicate a protracted decline between the Changhsingian and Olenekian, but the confidence bounds are too large for robust conclusions (Fig. 3). Compared to an earlier diversity analysis on the level of genera by Rees[7], our curve shows similar

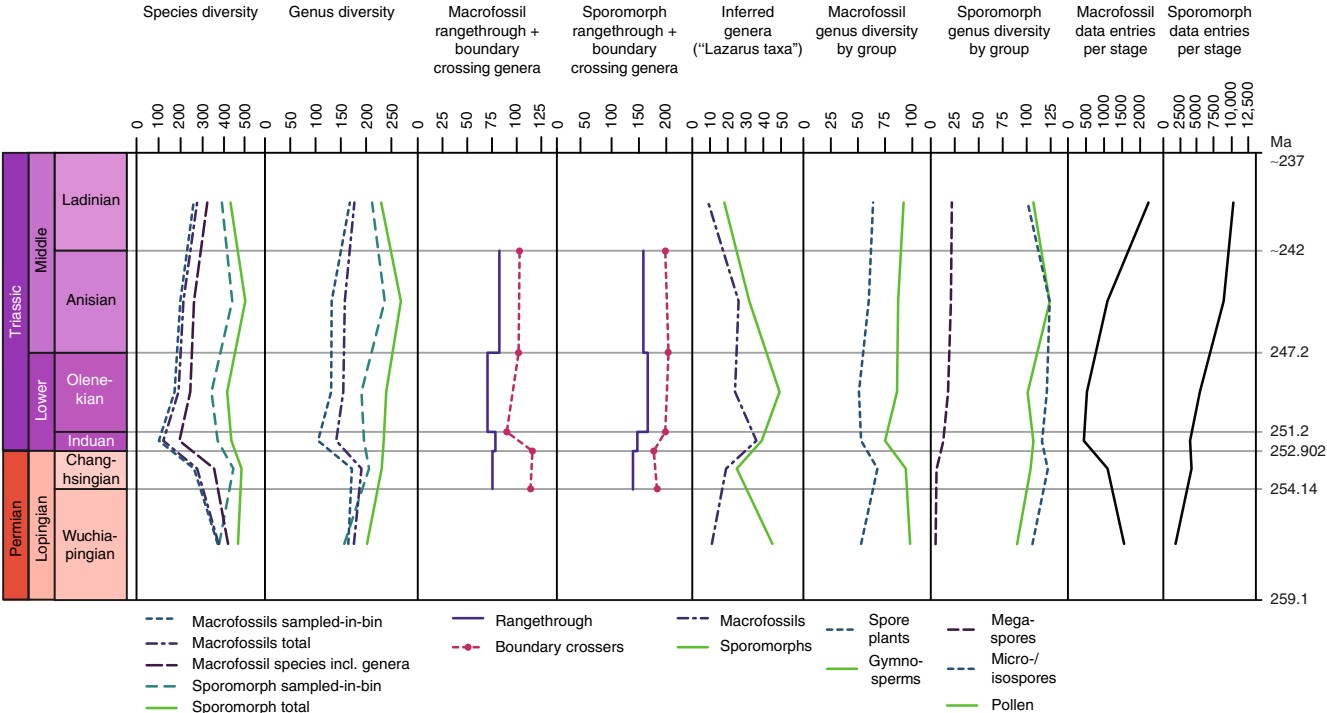

**Fig. 1** Diversity indices and distribution of data entries per stage. Results for sporomorphs and plant macrofossil taxa from the Wuchiapingian to the Ladinian

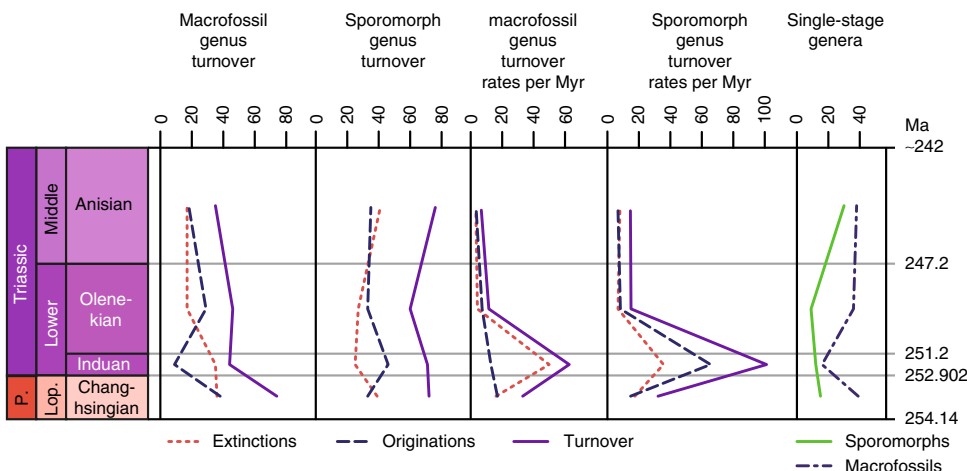

**Fig. 2** Origination, extinction, turnover and single-stage genera. Counts and rates per million years stage duration for plant macro- and microfossils. P. = Permian, Lop. = Lopingian

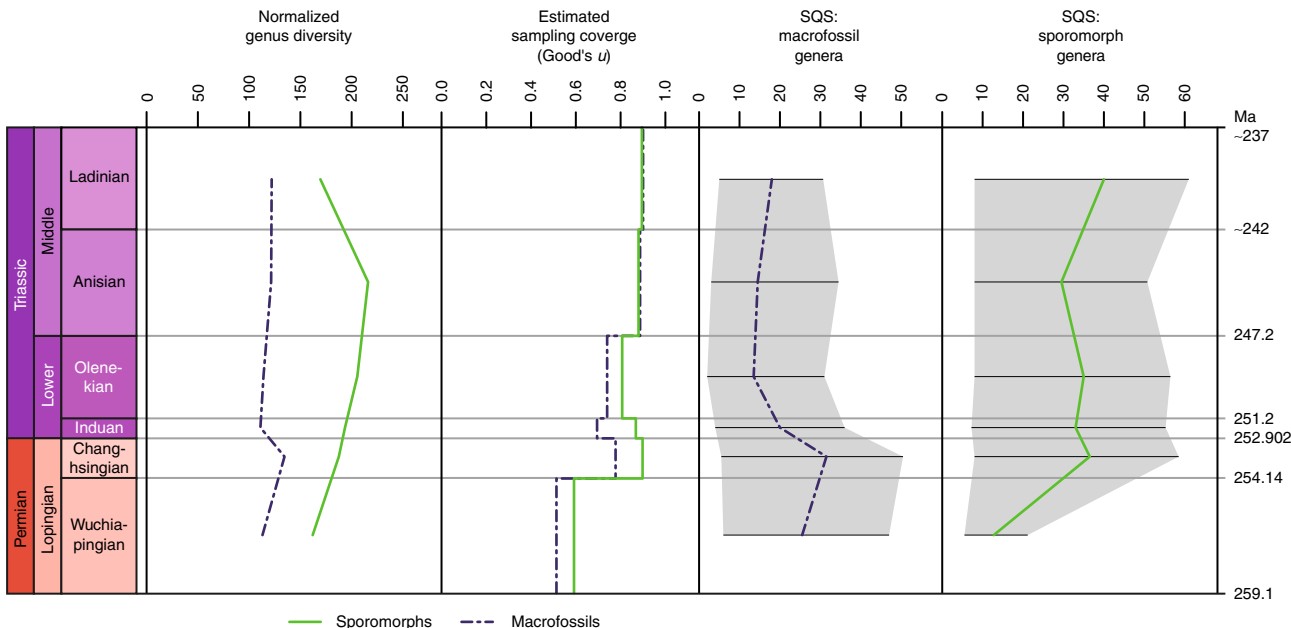

**Fig. 3** Normalized genus diversity, sampling coverage and shareholder quorum subsampling. Shareholder quorum subsampling (SQS) uses 'sqsbyref' function (with: quorum = 0.4, trials = 500, sample.quota = 5, merge.repeats = TRUE; see methods), showing median line and 95% confidence interval

trends, but with a mostly (except for the Wuchiapingian) higher diversity and less pronounced decline at the Permian–Triassic boundary (Figs. 4, 5a). The differences are explainable by the use of different sources, updated taxonomy and dating, while the two datasets clearly have a common underlying structure.

Plant macrofossil family diversity shows trends similar to genus diversity but with the peculiarity of a distinct decline between the Anisian and Ladinian, which contrasts with the trends seen in plant macrofossil species and genera, but agrees with the sporomorph records (see below). When comparing our family diversity curve with the recently published one by Cascales-Miñana and Cleal[5], the differences are striking (Figs. 4, 5b), even though the trends in our species curve would match perfectly with their families (Fig. 5c). The differences can be explained by a differing approach to partitioning the lower-level taxa into families (see Methods).

**Sporomorph record**. The diversity of sporomorph species and genera (Supplementary Data 1, 2) is generally higher compared to macrofossil diversity (Fig. 1), which is expected due to the generally higher preservation potential of sporomorphs, their transportability by wind and water even from distant habitats to favourable depositional settings, their sheer abundance and the possibility of a single plant species to produce multiple spore or pollen taxa. The Wuchiapingian record presents an exception with the number of sampled-in-bin species and genera being slightly lower for sporomorphs than for macrofossils. Both on species and genus level, the sporomorph record does not exhibit dramatic changes within the studied interval. Notably, it shows an increasing diversity from the Wuchiapingian to the Changhsingian and a decline from the Anisian to the Ladinian. On the species level, diversity also declines across the Permian–Triassic boundary and within the Lower Triassic, while diversity on the genus level remains more or less constant. In the Changhsingian, Olenkian and Anisian, the numbers of genera that originate and go extinct are almost balanced (Fig. 2). The Induan has a surplus of originations and the lowest number of extinctions, but relative to the short duration of the stage, extinction as well as origination rates are by far the highest within the studied interval. The

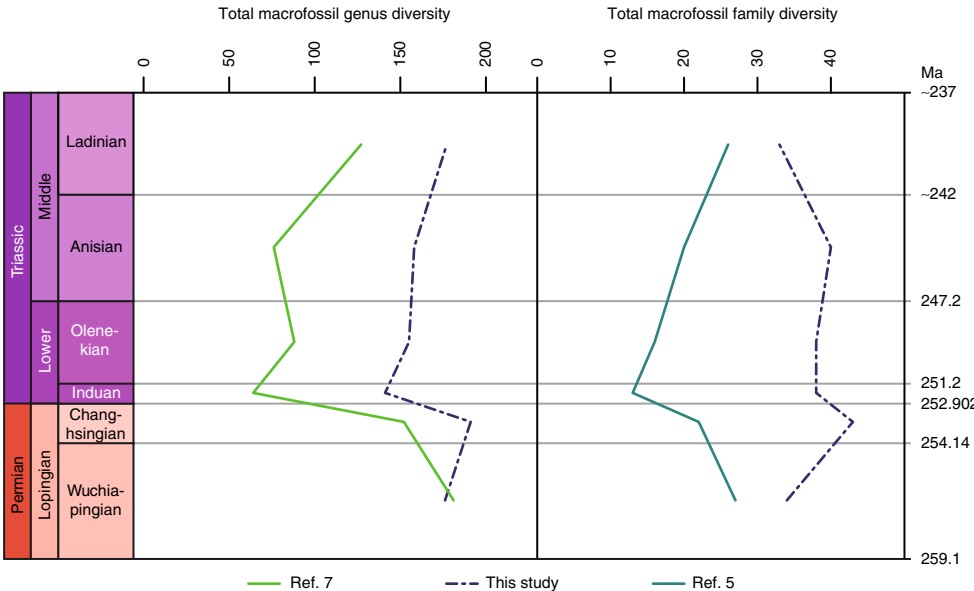

**Fig. 4** Comparison of macrofossil diversity presented in this study with previous works. With the exception of the Wuchiapingian, genus diversity in this study is higher than according to ref. [7], leading also to a less pronounced loss of diversity at the Permian–Triassic boundary, yet showing mostly similar trends. By comparison, family diversities in this study and ref. [5] are conspicuously different

Wuchiapingian is quantitatively under-represented in the data and has the lowest estimated sampling coverage (Figs. [1], [3], [5f]). The diversity of this stage is therefore probably underestimated, whereas the Ladinian is covered by the most comprehensive dataset. Shareholder quorum subsampling produced inconclusive results due to very large confidence intervals (Fig. [3]).

The biological affinity of specific sporomorph taxa can be determined if they are found in situ within macrofossils (sporangia or pollen sacs), but this is comparatively rare. The botanical affinity of most of the sporomorph taxa in this study is unknown. However, these sporomorphs can be assigned to three main categories; pollen, produced by gymnosperms, iso-/micro-spores, produced by bryophytes, lycophytes, pteridophytes and sphenophytes, and megaspores, derived from heterosporous lycophytes and pteridophytes. The diversity curves of spores and pollen show almost identical trends (Fig. [1]). Spore diversity is generally higher compared to pollen diversity, except for the Ladinian if megaspores are not counted. Megaspores have a negligible diversity in the Lopingian and diversify throughout the Lower and Middle Triassic.

**Plant groups.** Generic diversities of spores and spore plant (pteridophyte, lycophyte, sphenophyte and bryophyte) macrofossils show the same trends (Figs. [1], [5h]), except for the Ladinian. There, the diversity of macrofossils increases, whereas spore diversity declines. By contrast, the datasets of pollen and gymnosperms show completely different diversity trends (Figs. [1], [5i]). The gymnosperms show a more classical diversity curve with a declining trend in the Lopingian and across the Permian–Triassic boundary, followed by a recovery phase. However, in this case, the recovery seems to be mostly complete by the Olenekian. The Induan sticks out as an interval of considerably but only briefly lowered diversity. A comparison of all main plant groups shows that substantial diversity losses at the Permian–Triassic boundary are only recorded in pteridophytes, pteridospermatophytes and cycadophytes, while other plant groups are barely affected, with conifers and ginkgophytes even increasing in diversity (Fig. [6]).

**Potential biases.** There is a strong and significant correlation between the number of entries per stage and the (sampled-in-bin) genus and species diversities for macrofossils (both: Spearman's $r_s$ = 0.83, $p$ = 0.029; Supplementary Data 3), which may indicate an impact of sampling bias. By contrast, the correlation of the data distribution (number of data entries per stage) with sporomorph genus diversity ($r_s$ = 0.77, $p$ = 0.051) is less strong and just outside the common limit for statistical significance. It is negligible for sporomorph species ($r_s$ = −0.314, $p$ = 0.75). This does not imply that species are less affected by sampling bias but may point to other biases such as inconsistent taxonomy on the species level.

The low diversity of macrofossil taxa in the Induan may well be an artefact, as there are several reasons to consider the Induan under-represented, despite the considerable attention it has received. In the first place, quantitatively less data is available for the Induan and Olenkian than for the other stages (Figs. [1], [5e]). Both Induan and Olenkian have a lower estimated sampling coverage than the Changhsingian (Fig. [3]), which increases the relative probability that surviving taxa have not been discovered. Secondly, the total species diversity in the Induan is lower than the total diversity of genera, unless genera without a named species and inferred presences of long-ranging but unsampled genera are counted for species diversity (logically, at least one species per genus must have existed) (Fig. [1]). This indicates general quality issues of the fossil record from this stage. Thirdly, the proportion of macrofossil genera that are recorded from previous and following stages but show an intermittent absence in the Induan ("Lazarus taxa") is higher than for the other stages (Fig. [1]). Fourthly, there is no decrease in sporomorph genus diversity. Last but not least, the Induan is chronostratigraphically much shorter than the other ages (ca. 700 kyr for the Induan Age compared to 2.2 to 5.2 Myr for the Wuchiapingian, Changhsingian, Anisian, and Ladinian ages according to the latest version [2018/08] of the International Chronostratigraphic Chart[26]). Thus, the time window for plant fossil preservation was limited. If the Induan is indeed critically under-represented and its low diversity erroneous, this would imply that diversity is underestimated in the Changhsingian, Induan and to some extent in the Wuchiapingian and Olenekian due to the

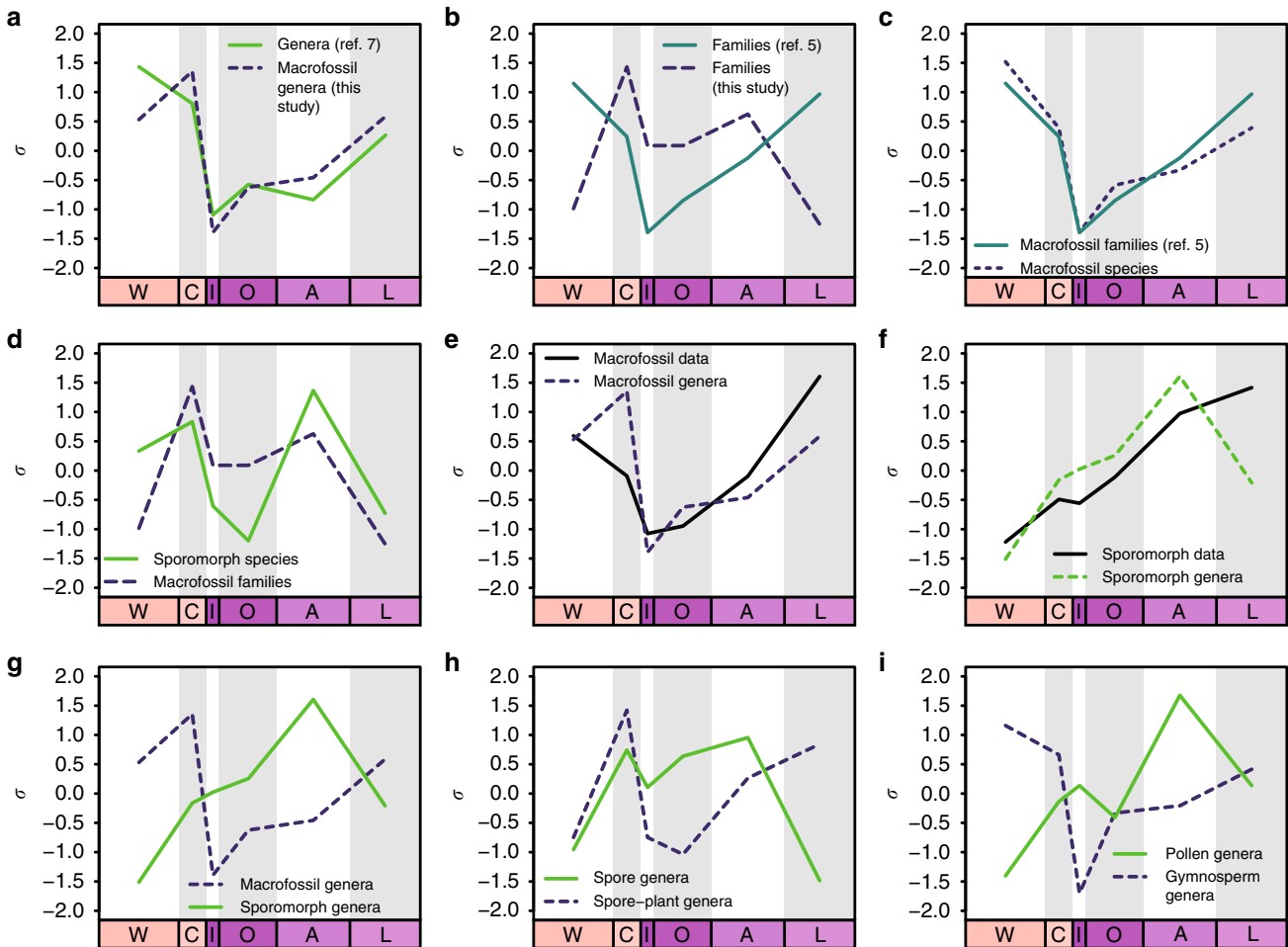

**Fig. 5** Comparison of results after mean-centreing and scaling to the standard deviation (σ). W = Wuchiapingian, C = Changhsingian, I = Induan, O = Olenekian, A = Anisian, L = Ladinian. **a** Total diversities of macrofossil genera in this study and ref. [7], showing similar trends, except across the Wuchiapingian to Changhsingian transition. **b** Plant families presented in the present study and in ref. [5], showing conspicuous differences. **c** Total diversities of macrofossil species presented in this study and plant families presented in ref. [5]; note that the curves are almost congruent, despite the differences at the family level shown in **b**. **d** Total diversities of sporomorph species and plant macrofossil families, exhibiting broadly similar trends. **e** Macrofossil data entries and macrofossil genera; note the similarity with the exception of a peak in generic diversity in the Changhsingian. **f** Microfossil data entries and microfossil genera; these are well-correlated except for a significant decline in the number of genera towards the Ladinian despite an abundance of data. **g** Plant macrofossil and microfossil genera, exhibiting very different trends. **h** Genera of spores and spore-plant macrofossils, showing several key similarities; compare with **i**. **i** Pollen and gymnosperm macrofossil genera, showing almost perfectly opposite trends; compare with **h**

Signor-Lipps[27] and Jaanusson[28] effects. Consequently, extinctions in the late Permian would be overestimated, whereas originations are underestimated in the Induan and overestimated in the Olenekian and/or the Anisian. In short, the observed fluctuations in macrofossil diversity appear to be mostly artifactual.

## Discussion

The term "mass extinction" is often used without a precise definition. However, such events have also been quantitatively identified as significant excursions from a linear regression of the extinction rates (per million years)[2,5]. This method is not applicable to our data due to the restricted temporal scale. Within the studied interval, the highest rates of extinction occur in the Induan in both sporomorphs and plant macrofossils, but an analysis of broader scale with the same methodology would be required to determine how these rates compare to the rest of the land plant fossil record. By contrast, even if the loss of diversity in land plant macrofossils at the Permian–Triassic boundary as presented herein is accepted as a true signal, its low magnitude (19% of Changhsingian macrofossil genera and 17% of

sporomorph genera going extinct; Supplementary Data 3) and selectivity do not justify calling it a mass extinction. A mass extinction of land plants in the latest Permian had been postulated based on observations in both single sections and global diversity curves derived from macrofossils. The comprehensive dataset presented herein does not support this interpretation. Of course, our dataset is ultimately not complete either. It is global, but not all regions and stages are equally well represented, which is a source for bias when considering the presence of provincialism, the possibility of migration and staggered extirpation. Another locally significant limitation on completeness is the fact that the taphonomic window for plant preservation is not equally close to all the habitats of various plant communities and transport may (selectively) destroy most plant remains in different sedimentary settings[29–33]. The climate in the late Permian and Early Triassic is thought to have been unusually hot, which would have affected the distribution of both plant habitats and areas with conditions favourable for fossilization[34]. Furthermore, different settings are sampled unequally. There is a bias towards sampling wetlands over drylands[35], with wetlands being particularly affected by the Permian–Triassic boundary event[17,19,36].

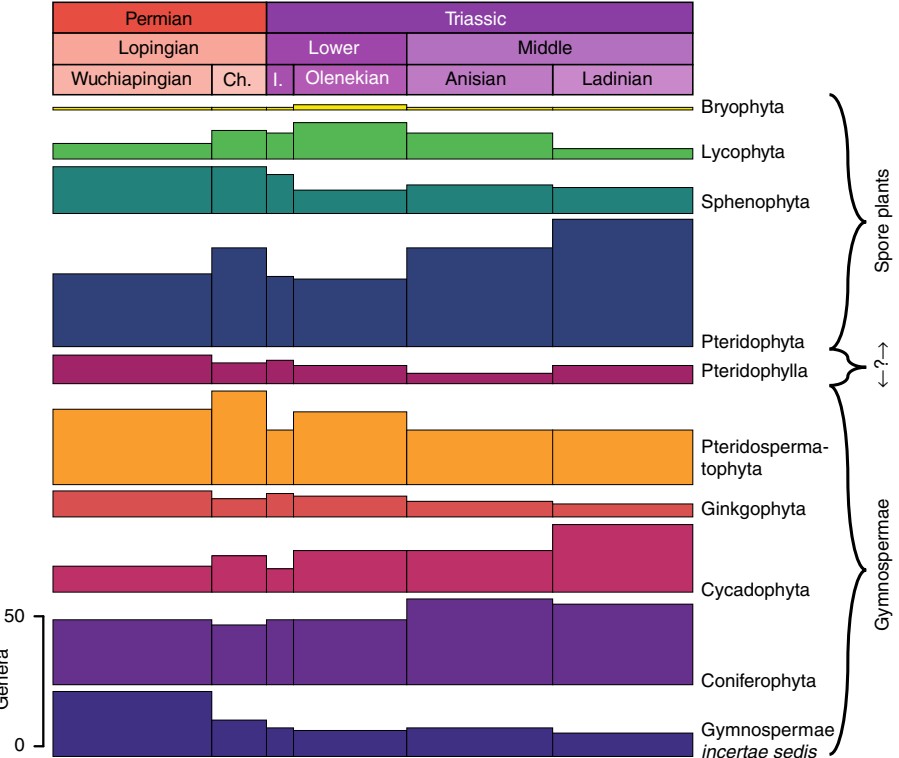

**Fig. 6** Sampled-in-bin genus diversities of macrofossils per main plant group. Divisions are grouped as spore-producers and gymnosperms, with Pteridophylla comprising uncertain cases. I. = Induan. Ch. = Changhsingian

In the current state, there is no convincing evidence for a global mass extinction among land plants at the end of the Permian. Considering previous studies, it appears that none of the major mass extinctions in the animal fossil record was mirrored by a mass extinction in plants[5,6,14,37,38]. The fossil record of land plants is marked by almost uninterrupted periods of diversification or relatively stable diversity. However, the compositions of floras changed repeatedly throughout the history of land plants. By all accounts, their dominance structures were also drastically altered during the Permian–Triassic transition both on the short and long term[13,17,37,38]. Furthermore, peat-forming habitats disappeared due to the absence of suitable plants or due to environmental conditions[14,19], a high abundance of undispersed spore tetrads and teratological pollen grains indicates a disturbance of the reproduction ability of the producing plants, possibly caused by intense UV-B radiation[21,39,40]. The organic-walled microfossil *Reduviasporonites*, which occurs abundantly near the Permian–Triassic boundary in many localities[41,42], has been interpreted as a fungus infecting the plants[43,44], but also as a saprophyte decomposing dead plants[45] and even as an alga[46–48]. While plant communities certainly reacted to the environmental disturbances that caused the end-Permian mass extinction, extinction rates were only considerably elevated in the Early Triassic, and the overall diversity loss limited. Plants may have survived even in small refugia[49], which would be unlikely to appear in the sampled fossil record. Further reasons why plants (individually or as a group) seem to be resilient with respect to the environmental hazards causing animal mass extinction events are their numerous autecological advantages. These include resistant and dispersable resting stages (spores and seeds may still be viable after decades[50–53]) in their life cycles, regenerative features and subterraneous structures[39,54], whereas most animals are susceptible to rapidly changing conditions. It should be noted that angiosperms, which dominate most terrestrial floras today, depend in their majority on interactions with certain animals for pollination and seed dispersal. Likewise, many modern herbivorous animals are adapted to a particular diet. Such mutualistic dependencies may have been in effect between certain insects and some late Palaeozoic pteridospermatophytes and conifers[55–57]. The insects involved died out in the end-Permian mass extinction, and modern forms of mutualism developed during the Mesozoic[55–57]. The loss of their animal partners does not guarantee the extinction of a plant species, but generally increases extinction risk[58]. In this way, the mass extinction among insects potentially contributed to the diversity loss in pteridospermatophytes at the Permian–Triassic boundary. A mass extinction among terrestrial animals today might also affect dependant plants—most importantly the dominant angiosperms[59]—and vice-versa. On the other hand, judging by their records, other major plant groups that rely on abiotic vectors would have a comfortable chance of survival.

## Methods

**Data collection and treatment**. Occurrence data (Supplementary Data 4, 5) was taken from literary sources and unpublished collections (Supplementary Note 1, Supplementary Table 1). The records were corrected for (reasonably recognizable) spelling errors and synonymies. Stratigraphic information from the original publications was updated if possible and correlated with the global stages according to the current international chronostratigraphic chart[28].

**Taxonomic concepts**. Macrofossils that are known to be biologically connected are treated as the same genus (whole-plant concept). Families are assigned mainly following ref. [60]. Orders without recognized families are treated as a single family. By comparison, ref. [5] follows the family concept of ref. [61], which is generally more restrictive and involves several families representing a single genus in our approach.

**Data selection**. Only records with a sufficiently constrained date were used for diversity calculations. We excluded occurrences of sporomorph taxa from the calculations if the authors of the source material considered them as reworked or possibly reworked. For species-level analyses, records in open nomenclature ("sp.", "spp.", "cf.", "aff.", "?", etc.) are excluded. For genus-level analyses, records of species that are only tentatively assigned to a genus are excluded. Calculations (see below) were based on 8327 entries on macrofossils (Supplementary Data 4) and

34206 entries on spores and pollen assigned to the Wuchiapingian to Ladinian stages (Supplementary Data 5). Additionally, occurrences that pre- or post-date the studied interval (222 and 1078 entries respectively on macrofossils, 1075 and 891 entries on spores and pollen) were used to extend taxon ranges, but were not entered systematically.

**Calculations.** Calculations were performed using custom code (Supplementary Data 6) in R (version 3.3.2)[62], employing the 'sqsbyref' function of John Alroy (available online: http://bio.mq.edu.au/~jalroy/sqsbyref.R; accessed 12 November 2018). Results are presented in Supplementary Data 1–3.

Four basic types of taxa present in a particular interval are distinguished: Taxa that are present only in this single interval; taxa with ranges crossing the lower or bottom boundary of the interval, but not the upper or top boundary (i.e., going extinct); taxa that first appear in this interval and cross the top boundary (i.e., originating); taxa that are present before and after the interval (range-through taxa)[63]. The number of range-through taxa may either be counted by considering only documented occurrences (occurrence method[64]) or by inferring their presence between the first and last occurrence even in intervals from which they have not been reported so far and thus appear as Lazarus taxa (range method[64] or range-through method[65]).

Sampled-in-bin diversity is obtained by counting the number of taxa reported from a specific stage (occurrence method).

Total diversity refers to the number of taxa that are either reported from a specific stratigraphic unit or inferred to be present from both older and younger records. This relates to the range-through method and is also referred to as range-through diversity.

Normalized diversity is used to estimate the mean standing diversity of each stage irrespective of its duration by counting range-through taxa plus half of the taxa originating and/or going extinct within the stage[66].

Origination, extinction, and turnover (origination + extinction) are here calculated excluding taxa appearing in only one stage, i.e. by counting bottom and/or top boundary crossers. Origination, extinction and turnover rates are calculated by dividing the raw counts of origination and/or extinction events by the respective stage duration in Myr according to the latest ICS chart[26].

Shareholder quorum subsampling (SQS, also known as 'coverage-based rarefaction') is a method to counter unequal sampling between intervals by randomly drawing from the pool of occurrences, samples and references until a certain quorum of frequency coverage is reached[67–69]. Coverage is estimated through a modified formula proposed by Good[70] based on the ratio of the number of taxa with only a single occurrence ($n_1$) to the total number of occurrences ($N$) considered:

$$u = 1 - n_1/N \tag{1}$$

In the version of SQS employed in this study, references are drawn randomly without replacement and the occurrences of up to five samples of that reference are added to the selection on which coverage and the subsampled taxon richness are calculated. The coverage is calculated after each draw, with occurrences of a particular taxon in multiple samples of a single reference binned as one (option merge.repeats = TRUE). This prevents the overestimation of coverage regarding references reporting a small number of taxa repeatedly. The taxon richness is recorded each time when the coverage reaches or overtakes the chosen quorum of 0.4. The whole process is repeated 500 times and the median value along with the lower and upper bounds of the 95% confidence interval of all recorded taxon richness counts at the quorum level or above are returned.

Data distribution is calculated as the number of entries per stage in the raw data. An entry can relate to a single specimen or multiple specimens, but is here treated as a single occurrence. In each case, the number of entries corresponds to the resolution of sampling effort in the source reference.

Correlations are tested for using Spearman's rank correlation coefficient $r_s$ or $\rho$ (rho), which tests for a monotonous relationship between variables (Supplementary Data 3). A correlation is considered strong if $r_s < -0.6$ or $r_s > 0.6$, and statistically significant if the probability of the null hypothesis $p < 0.05$. Both $r_s$ and $p$ are calculated with the 'cor.test' function in R.

In order to better display concurrence of trends, selected pairs of indices (Fig. 3) were centred by subtracting their respective mean and scaled to their standard deviation ($1\sigma$).

**Code availability.** Code to reproduce the presented results is provided in Supplementary Data 6.

**Reporting summary.** Further information on experimental design is available in the Nature Research Reporting Summary linked to this article.

## Data availability
References of publications used as data sources are listed in the Supplementary References (see also Supplementary Note 1). The aggregated database is provided in Supplementary Data 4 and 5. A reporting summary for this article is available as a Supplementary Information file.

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

## Acknowledgements

This work was funded by the Euregio Science Fund (call 2014, IPN16) of the Europaregion/Euregio Tirol-Südtirol-Trentino/Tirolo-Alto Adige-Trentino. A visit of H.N. at the Natural History Museum in London was supported by SYNTHESYS (Access Call 4, 2016; GB-TAF-6751). Anna Fijałkowska-Mader (Geological Museum in Kielce, Polish Geological Institute), Eugeny Karasev (Borissiak Paleontological Institute, Moscow), and Carmen Heunisch (LBEG Niedersachsen, Hannover) provided literature. Matthias Franz (Universität Göttingen), Eugeny Karasev, Stephen McLoughlin (Naturhistoriska riksmuseet, Stockholm) and others helped with stratigraphical correlations. John Alroy (Macquarie University) provided the code for shareholder quorum subsampling.

## Author Contributions

H.N., E.S.-H. and E.K. conceived the paper and contributed data. H.N. analysed the data, and wrote the manuscript. All authors revised the manuscript.

## Additional information

**Competing interests:** The authors declare no competing interests.

