## [Peer Review File · Nature Communications]

Reviewers' Comments:

Reviewer #1:

Remarks to the Author:

Review of Nowak et al.: A non-extinction event for plants during the end-Permian mass extinction

The authors have undertaken a global assessment of the more limited macrofloral and more abundant palynological record in the final two stages of the Permian and first four stages of the Triassic, to test the prevailing extinction paradigm that terrestrial plants experienced a similar reduction in diversity as that recorded in the marine realm. Using previously published works and unpublished data sets, the authors examine a myriad of assemblage data and conclude that terrestrial plants, neither at the family, genus, or species level, did not experience any diversity demise. These results are in accord with previously published work of Schneebeli-Hermann and collaborators who have examined palynological records on a regional basis. This represents another, but more comprehensive, compilation and data analysis, the first having been published by Rees (2002), from which the prevailing end-Permian paradigm is demonstrated to require revision. The manuscript is well written, easy to follow, and effective. It neither exaggerates the significance of their findings, nor claims to provide an analysis on a limited data set. In fact, the data set includes 7358 macrofloral and 42709 microfloral records which are binned to the stage level. As such, the results pertain to this level of chronostratigraphic resolution and demonstrate the overall patterns in the data set.

Using Ziegler's compilations that formed the basis of his Paleogeography and Paleogeographic Atlas Project, Rees et al. (2002) examined the relationship between Permian-aged macrofloral assemblages and climatically sensitive sediments. Results from this project led Rees (2002) to evaluate the macrofloral records across the Permian–Triassic wherein he reached a similar conclusion as the current study. Yet, Rees' data set is only at the generic level and encompasses 561 genera in the six-stage bins examined in the current analysis. The omission of genera of dubious systematics reduces the number in Nowak et al.'s analysis to 402 genera. But, the current study extends Rees' analysis to the species level at the macrofloral level, and adds both generic and species analysis of the palynological record, enhancing the patterns previously recognized. The current contribution, then, not only replicates Rees' pattern, but extends it with a more comprehensive data set. Rees declined to use species-level macrofloral records due to their uncertainty in many instances; the current study has used a protocol to eliminate the problem. Hence, the results of Nowak et al.'s contribution are of interest not only to the paleobotanical/paleontological community, but to the wider biological community, at large.

This reviewer is relieved to see that these authors have taken a non-parametric approach in their data analysis, in contrast to studies that rely on parametric statistical analyses when normally distributed data sets are not demonstrated. The Excel spreadsheets provide for all raw data and data matrices in the supplemental materials. The only thing missing in the latter is a color key for the cells in 16829_0_data_set_2970257_p6bjm.xls. Therefore, a colored key must be included to explain cell colors for the Spearman rho results, and their p-values, as the same colors are used for each matrix. Once the p-value matrix is studied, it becomes clear that green cells are those that are statistically significant whereas red cells indicate pair-wise comparisons that are not statistically significant. Yet, dark orange and light orange/red colors also indicate insignificant correlations, but at differing p-values. Either a pair-wise analysis is or is not significant, leading this reviewer to wonder why there is a range of colors in the data matrix. A pair-wise comparison that is "just outside" the level of significance is still insignificant. If the authors have one or more valid reason(s) for justifying the color gradient, it is unstated and, in this reviewer's mind, unnecessary.

The authors provide no reference or details on the methods employed wherein unnamed "selected

indices" were centered (using which statistical method) and scaled to their standard deviation (unstated as to whether the scaling is to 1 sigma or 2 sigma). More detail is needed to demonstrate to the reader that the centering and scaling in R followed established protocols, and wasn't a parameter chosen in the program "just because" it was available in the software. What do the data plotted in Figures 3g & 3h mean when, for example, macrofloral data entries (N=?) are plotted against macrofloral general (N=?), and sporomorph data (N=?) are plotted against sporomorph genera (N=)? These plots require clarification for the reader. Similarly, reasons as to why the Cacaes-Miñana and Cleal data set is used for comparative purposes rather than Rees' supplemental data or the Paleobiology DataBase are not provided, and should be addressed at least in the supplemental documentation.

The conclusions drawn by the authors are robust, valid (as whatever validity is limited by the available dataset and whatever inherent biases might affect it), and repeatable. Hence, the repeatability of the project, with transparency in the supplemental materials, makes the conclusions reliable.

Suggested improvements:

- Improvements can be made to the colors used in the figures. For example, standard colors to indicate Permian (CYMK 5/75/75/0 and colors for epochs and stages) and Triassic (50/80/0/0 and colors for respective epochs and stages) records are used in all figures which is appropriate. Unfortunately, the Triassic's purple colors do not provide sufficient contrast with the overlying text. That text should be changed to a white font for ease of read.
- Similarly, the dark gray bands in Fig. 3 used to help the reader follow trends across stages should be lightened.

Does this manuscript reference previous literature appropriately? If not, what references should be included or excluded?

A more appropriate and pertinent reference to Looy et al. (2014) is the comprehensive review of DiMichele, W.A., and Gastaldo, R.A., 2008, Plant Paleocology in Deep Time: Annals of the Missouri Botanical Gardens, v. 95, no. 1, p. 144-198.

The abstract is clear and accessible, and the introduction and conclusions appropriate. Evidence continues to be published, though, that the paradigm of an end-Permian extinction event of terrestrial vertebrates is not coincident or coeval with the marine crisis, nor may there even be a turnover in fauna.

Reviewer #2:

Remarks to the Author:

I believe this to be a very important manuscript that is most definitely worthy of publication in a Nature journal. The paper should attract the attention of a broad range of scientists, and of the public. It includes an excellent empirical data set that is used to examine a hypothesis of long standing regarding the response of terrestrial plants to the events that occurred near the Permian-Triassic boundary.

For many years there has been, in my opinion, an untenable acceptance of a terrestrial-plant mass extinction at the Permian-Triassic boundary, paralleling that in the reasonably well documented marine-animal record. This belief was based on terribly weak data. It actually does not comport with the survival of nearly all major clades through the supposed extinction interval, indicated by the fact that, although absent in the Early Triassic, they make their reappearance well afterward, during the Middle or Late Triassic. Certainly, these reappearances do not represent massive evolutionary

convergence across many lines of the phylogenetic tree. The groups were "out there" somewhere, surviving beyond the preservational window. Yet, as the old saying goes, "possession is nine tenths of the law"; so once this mass-extinction belief became entrenched (and who doesn't love a mass extinction?), it also became nearly impossible to displace. Yet, over the past several years I have had many informal discussions with a variety of scientists who work on terrestrial fossils, all expressing grave doubts about a terrestrial extinction – in many instances, not so much that it did not occur, but that the terrestrial record, as presently constituted, is too poor to provide a definitive answer.

Here, I believe, the authors present a data set that addresses these matters squarely. Pollen and spores, unlike any other terrestrial fossils, plant or animal, have a high preservation potential, are extremely widely dispersed, and most often are drawn from broad areas of the terrestrial landscape. They require a lot of specialist training to interpret, which is why such data have not been synthesized in much of the geological record (with notable exceptions). The authors, furthermore, make their case without hyperbole and without engaging in the nastiness that seems to be part of much of the Permian-Triassic boundary literature.

It is my personal experience with the terrestrial plant macrofossil record that it has significant biases. These biases, expressed in several papers (I would point to papers in the plant taphonomy literature by Burnham, Gastaldo, Pfefferkorn, Falcon-Lang, and others) and in my own writing (the authors cite a recent paper by Looy et al.), indicate clearly that there are often enormous discontinuities/gaps in the distribution of taxa, often at the genus and species levels of resolution, amounting from millions to 10s of millions of years long. There are now many reports in the literature not only of so-called "Lazarus" taxa (reappearances long after the organisms had been thought to be extinct), but also well before they were thought to have originated (what we've dubbed "Methuselah taxa"). These kinds of patterns do not mean the terrestrial record cannot be used to address interesting problems in biology. What they indicate is that the terrestrial record must be used judiciously and in ways commensurate with its strengths and weaknesses. – I believe this paper does just that.

Detailed comments are keyed to line number. They are primarily editorial. I find little to disagree with in the science presented here. This is about as "tight" a study as one might ever hope for.

Line 10: delete "following"

Lines 15-16: I suggest rewording "the taxonomic...plants)" to say "the taxonomic diversities of gymnospermous macrofossils and of the pollen they produced"

Line 17: It is unclear from the wording here if gymnosperm macrofossils are undersampled, or if all gymnosperm fossils, macro and micro, are undersampled. This should be clarified because a distinction has been made between these earlier in the paragraph.

Line 33: Change "On the southern hemisphere" to "In the Southern Hemisphere"

Line 37: Change "account for" to "be accounted for by"

Line 43: Change: "unused, as in most cases" to "unused and, as in most cases,"

Line 46: I suggest changing "has been attempted" to "is presented here". "Has been attempted" is a very equivocal statement, and carries with it an implication of having tried and failed. If some equivocation is necessary, then say that the study "is presented here, based on presently available data".

Line 56: Is this the first use of the acronym "PTB"? Perhaps it should be first used parenthetically in line 53, following the words "Permian-Triassic boundary".

Line 64: Change "which contrasts the" to "which contrasts with the"

Line 69: It would be better to say "approach to partitioning" instead of "approach in partitioning".

Lines 73-74. I might add here that sporomorph diversity is higher than macrofossil diversity because of the transport potential of spores and pollen, by both wind and water, which draws in elements from a wider diversity of habitats than macrofossils, and also has the potential to sample the regional flora. These are very important points.

Line 87: change "unknown, however," to "unknown. However,"

Lines 88-90. rewrite

Replace the semi-colon after categories with a colon.

Delete "are" after "pollen" and replace it with a comma

Delete "are" after "iso-/microspores" and replace with a comma

Delete "are" after "megaspores" and replace with a comma

Line 96: This is just a suggestion, here and elsewhere in the manuscript. Because of the complexities of botanical terminology, and although "everybody knows what you mean", it is technically incorrect to call non-seed plants "spore plants" (seed plants still produce spores, we just have specialized names for them). It would be better to refer to the "spore plants" as "pteridophytes" or "pteridophytes and bryophytes" (or even "pteridophytes, bryophytes, and fungi")

Line 101: I suggest replacing "already in" with "by"

Lines 110-113. It actually is not clear to me from the titles of the Excel data sheets which one of these is Table 4, but it would seem to be the one with all the correlations in it.

That said, in lines 110 & 111 it is stated that there is a strong correlation between "the number of entries per stage" and "genus diversity". OK, that is very clearly stated, and understood without having to look at the Table for myself. In lines 112 & 113, however, it is stated that there is a less strong correlation "of the data distribution" with "macrofossil species diversity" and with "sporomorph genus diversity". I cannot follow/understand this statement, or interpret it from, what I presume to be, Table 4. What is "the data distribution"? Number of samples per time bin? This is crucial and must be made more clear; if it is sample number per time bin, then I suggest using the same language as above: "the number of entries per stage". If it is something else, then clarification is needed.

Line 120: Change "least" to "lowest"

Line 121: Change "are not" to "have not been"

Lines 141-142: I suggest putting a comma after "signal", changing "does" to "do", and putting "mass extinction" in quotation marks.

Line 194: Change "constraint dating" to "constrained date"

Line 202: "total diversity", as it is used in this sentence, is often referred to as "range through" diversity, because ranges of taxa are extended through intervals where their actual occurrence has

not been documented. I do not think "total diversity" is a term that is generally or systematically used interchangeably with range-through diversity.

End of comments

William A. DiMichele
Department of Paleobiology
NMNH Smithsonian Institution
Washington, DC 20560
dimichel@si.edu

Reviewer #3:

Remarks to the Author:

This paper revisits changes in plant diversity across the PT mass extinction, proposing provocatively that the extinction had almost no substantial impact on plants. This is a paper of potential broad interest, and with important implications. Unfortunately, the validity and significance of the work cannot be assessed because the underlying data has not been made available. Moreover, the taxon-counting approach used does not take into account the severe biases that exist in the fossil record and the results might be flawed as a result, and there are also problems with the statistical analyses.

My number one concern is that the dataset underlying the study, which is arguably the most significant part of the paper, is not being made available to other researchers. The methods sections states that the database is available from the corresponding author "upon reasonable request". This is unacceptable, as it makes the research unrepeatable - who decides what a "reasonable" request is, and what happens if the corresponding author leaves the research field or is no longer contactable. Any acceptance of this manuscript should be contingent on the raw data being made immediately available to readers, in line with current standards within the research field, as well as my reading of journal policy:

"Authors are required to make materials, data, code, and associated protocols promptly available to readers without undue qualifications."

I also have a variety of other comments, detailed below:

- The methods consist of raw taxon counts, but there is a vast literature stretching back >40 years that has shown beyond all reasonable doubt that the fossil record is highly biased and that raw, observed counts of taxa are highly likely to be misleading. An array of different statistical approaches - rarefaction, shareholder-quorum subsampling, diversity residuals, TRiPS, phylogenetic diversity estimates etc. - have been developed to deal with this problem, and try to accurately estimate diversity change. This manuscript does not use any of these methods, but there is no discussion of why not, and why the plant fossil record apparently can be read at face value. The authors should either engage with these methods, or provide explicit justification for their approach.
- The authors repeatedly refer to certain stages being 'quantitatively underestimated' or having poor 'coverage', yet no information is provided on how this is defined. 'Coverage' is a term with a precise meaning in diversity analyses (see papers by John Alroy on SQS), but I do not think the authors are using the term with the same meaning.

- The authors do not explain how they calculate extinction and origination rates. They also do not really explain why greatly elevated macrofloral extinction rates in the Induan do not correspond to a mass extinction.
- Genus diversity is compared to 'number of entries' per stage to assess sampling biases, but it is completely unclear what 'number of entries' means, in part because the data is not available.
- The normal threshold for statistical significance is $\alpha = 0.05$. However, the authors refer to $p = 0.058$ as "significant", and still apparently consider $p = 0.14$ as significant, albeit "less significant", whereas this would definitely be considered non-significant by other researchers. Clarification of the stats is required.
- Comparisons of time series should use approaches that account for serial correlation (e.g. generalised least squares), which can inflate correlation coefficients. The authors should apply such approaches for their comparisons of "entries per stage" and diversity.
- "Lazarus" taxa are discussed but their abundance does not appear to be shown in any of the plots.
- Changes in diversity are often referred to as 'not catastrophic', 'slight' etc., but with no quantification of what these terms mean. Percentage diversity changes should be provided, and terms should be qualified. What do you mean by 'catastrophic'? What is your definition of a mass extinction, and why don't the diversity changes observed here comprise one? Explicit, quantitative discussion of this is required.
- Figures are not sufficiently often referred to in the text, and so it can be difficult to tell which figure should be examined (e.g. no figure references at all in lines 85-93).

Reviewer #4:

Remarks to the Author:

This is a very important paper that uses large and new data sets to analyze what happened to land plants at the Permian-Triassic transition. These results are highly significant because they show that plants reacted differently from land animals and marine invertebrates. These results correct some earlier studies that were based on limited data sets or the use of unsuitable taxonomic levels in plants.

Suggested title: No mass extinction for plants at the Permian-Triassic transition

The last sentence in the abstract beginning with "This means that none ..." is an overstatement and should be eliminated. This is not a conclusion that can be drawn from this paper.

Lines 37-38 should be changed to: ..., but it could also be accounted for by a severe taphonomic bias.

Lines 66-68: the paper by Cascales-Miñana & Cleal uses families and singletons and their conclusions are wrong. There is no mass extinction at the Carboniferous-Permian boundary and what they say about the P-Tr boundary is wrong, too. The most stable taxonomic level for plant macrofossils is the genus. The attribution to families depends on school and is often still artificial. There are also genera not yet attributable to any family.

Line 171: From the studies of Conrad Labandeira and his colleagues we know that insects and plants

had mutualistic relationships back in the Pennsylvanian. Are we certain that this was not also the case for at least some vertebrates? The phrase "Such mutualistic dependencies .. " is a strong statement that might not be correct and has no function here.

In Figures 1 and 2 the same scale should be used in each graph wherever possible. It is clear that it cannot be done in all cases.

Line 370: period and space missing before g

This manuscript demonstrates that several "things" happened to plants at the P-Tr transition but it was not a mass extinction. That is the important conclusion. The "things" that happened to plants were very well summarized in the publication by McElwain & Punyasena (2007). Perhaps the other aspect of the "not mass extinction" should be discussed more to give a more rounded account.

Response to reviewers' comments:

Reviewer #1 (Remarks to the Author):

Review of Nowak et al.: A non-extinction event for plants during the end-Permian mass extinction

The authors have undertaken a global assessment of the more limited macrofloral and more abundant palynological record in the final two stages of the Permian and first four stages of the Triassic, to test the prevailing extinction paradigm that terrestrial plants experienced a similar reduction in diversity as that recorded in the marine realm. Using previously published works and unpublished data sets, the authors examine a myriad of assemblage data and conclude that terrestrial plants, neither at the family, genus, or species level, did not experience any diversity demise. These results are in accord with previously published work of Schneebeli-Hermann and collaborators who have examined palynological records on a regional basis. This represents another, but more comprehensive, compilation and data analysis, the first having been published by Rees (2002), from which the prevailing end-Permian paradigm is demonstrated to require revision. The manuscript is well written, easy to follow, and effective. It neither exaggerates the significance of their findings, nor claims to provide an analysis on a limited data set. In fact, the data set includes 7358 macrofloral and 42709 microfloral records which are binned to the stage level. As such, the results pertain to this level of chronostratigraphic resolution and demonstrate the overall patterns in the data set.

Using Ziegler's compilations that formed the basis of his Paleogeography and Paleogeographic Atlas Project, Rees et al. (2002) examined the relationship between Permian-aged macrofloral assemblages and climatically sensitive sediments. Results from this project led Rees (2002) to evaluate the macrofloral records across the Permian–Triassic wherein he reached a similar conclusion as the current study. Yet, Rees' data set is only at the generic level and encompasses 561 genera in the six-stage bins examined in the current analysis. The omission of genera of dubious systematics reduces the number in Nowak et al.'s analysis to 402 genera. But, the current study extends Rees' analysis to the species level at the macrofloral level, and adds both generic and species analysis of the palynological record, enhancing the patterns previously recognized. The current contribution, then, not only replicates Rees' pattern, but extends it with a more comprehensive data set. Rees

declined to use species-level macrofloral records due to their uncertainty in many instances; the current study has used a protocol to eliminate the problem. Hence, the results of Nowak et al.'s contribution are of interest not only to the paleobotanical/paleontological community, but to the wider biological community, at large.

This reviewer is relieved to see that these authors have taken a non-parametric approach in their data analysis, in contrast to studies that rely on parametric statistical analyses when normally distributed data sets are not demonstrated. The Excel spreadsheets provide for all raw data and data matrices in the supplemental materials. The only thing missing in the latter is a color key for the cells in 16829_0_data_set_2970257_p6bjm.xls. Therefore, a colored key must be included to explain cell colors for the Spearman rho results, and their p-values, as the same colors are used for each matrix. Once the p-value matrix is studied, it becomes clear that green cells are those that are statistically significant whereas red cells indicate pair-wise comparisons that are not statistically significant. Yet, dark orange and light orange/red colors also indicate insignificant correlations, but at differing p-values. Either a pair-wise analysis is or is not significant, leading this reviewer to wonder why there is a range of colors in the data matrix. A pair-wise comparison that is "just outside" the level of significance is still insignificant. If the authors have one or more valid

reason(s) for justifying the color gradient, it is unstated and, in this reviewer's mind, unnecessary.
[emphasis added]

→ **The range of colors was used because it is not entirely clear if the common, yet not definitive threshold for statistical significance of $p = 0.05$ is the most appropriate in this case (see also below). However, for clarity, the color scheme was simplified and a legend was added.**

The authors provide no reference or details on the methods employed wherein unnamed “selected indices” were centered (using which statistical method) and scaled to their standard deviation (unstated as to whether the scaling is to 1 sigma or 2 sigma).

→ **The text was revised to clarify that the indices in question are those shown in Fig. 5 (corresponding to the former Fig. 3) and that centering is based on the mean. The y-axis labels in Fig. 5 were also changed to “ σ ”, and the text of the methods and the figure now clarify that the standard deviation equals 1σ : “[...] selected pairs of indices (Fig. 5) were centred by subtracting their respective mean and scaled to their standard deviation (1σ).” (lines 307-308); “Fig. 5. Comparison of results after mean-centering and scaling to the standard deviation (σ).” (line 574).**

More detail is needed to demonstrate to the reader that the centering and scaling in R followed established protocols, and wasn't a parameter chosen in the program “just because” it was available in the software.

→ **As mentioned above, more detail is now provided in the text. It should be noted that the centering and scaling is not used as a parameter for statistical analysis, but to facilitate visual comparison between trends in various curves.**

What do the data plotted in Figures 3g & 3h mean when, for example, macrofloral data entries ($N=?$) are plotted against macrofloral general ($N=?$), and sporomorph data ($N=?$) are plotted against sporomorph genera ($N=?$)? These plots require clarification for the reader.

→ **The plots (now in Fig. 5) are meant to show differences and similarities in diversity trends irrespective of absolute numbers, which are not as clear otherwise and are not entirely captured by numerical correlation analysis. The figure caption was extended to point out the relevant features (lines 576-589).**

Similarly, reasons as to why the Cacaes-Miñana and Cleal data set is used for comparative purposes rather than Rees' supplemental data or the Paleobiology DataBase are not provided, and should be addressed at least in the supplemental documentation.

→ **Data from the Paleobiology DataBase was used by Silvestro et al.²⁰⁹, and as such is also incorporated in our data. Since this represents only a small subset, a comparison seemed unnecessary. By contrast, the work of Cascales-Miñana and Cleal⁵ is conceptually similar, but independent from ours (although naturally the data sources overlap), and therefore provides a benchmark for the reproducibility of observed diversity patterns. The same is true of Rees^{6,7} genus level results. The latter are consequently now used for comparison as well (Fig. 4, lines 66-70).**

The conclusions drawn by the authors are robust, valid (as whatever validity is limited by the available dataset and whatever inherent biases might affect it), and repeatable. Hence, the repeatability of the project, with transparency in the supplemental materials, makes the conclusions reliable.

Suggested improvements:

- Improvements can be made to the colors used in the figures. For example, standard colors to indicate Permian (CMYK 5/75/75/0 and colors for epochs and stages) and Triassic (50/80/0/0 and colors for respective epochs and stages) records are used in all figures which is appropriate. Unfortunately, the Triassic's purple colors do not provide sufficient contrast with the overlying text. That text should be changed to a white font for ease of read.

→ **The color of the text has been changed to white for “Triassic”, “Lower” (Triassic), “Induan” and “Olenekian”.**

- Similarly, the dark gray bands in Fig. 3 used to help the reader follow trends across stages should be lightened.

→ **As suggested, we changed the hue. The corresponding figure is now Fig. 5.**

Does this manuscript reference previous literature appropriately? If not, what references should be included or excluded?

A more appropriate and pertinent reference to Looy et al. (2014) is the comprehensive review of DiMichele, W.A., and Gastaldo, R.A., 2008, Plant Paleocology in Deep Time: Annals of the Missouri Botanical Gardens, v. 95, no. 1, p. 144-198.

→ **Reference changed as suggested (lines 393-394).**

The abstract is clear and accessible, and the introduction and conclusions appropriate. Evidence continues to be published, though, that the paradigm of an end-Permian extinction event of terrestrial vertebrates is not coincident or coeval with the marine crisis, nor may there even be a turnover in fauna.

→ **We added a sentence on this topic in the introduction, with corresponding citations: “However, it has been questioned whether the terrestrial and marine events are coeval⁸⁻¹³,” (lines 28-29).**

Reviewer #2 (Remarks to the Author):

I believe this to be a very important manuscript that is most definitely worthy of publication in a Nature journal. The paper should attract the attention of a broad range of scientists, and of the public. It includes an excellent empirical data set that is used to examine a hypothesis of long standing regarding the response of terrestrial plants to the events that occurred near the Permian-Triassic boundary.

For many years there has been, in my opinion, an untenable acceptance of a terrestrial-plant mass extinction at the Permian-Triassic boundary, paralleling that in the reasonably well documented marine-animal record. This belief was based on terribly weak data. It actually does not comport with the survival of nearly all major clades through the supposed extinction interval, indicated by

the fact that, although absent in the Early Triassic, they make their reappearance well afterward, during the Middle or Late Triassic. Certainly, these reappearances do not represent massive evolutionary convergence across many lines of the phylogenetic tree. The groups were “out there” somewhere, surviving beyond the preservational window. Yet, as the old saying goes, “possession is nine tenths of the law”; so once this mass-extinction belief became entrenched (and who doesn’t love a mass extinction?), it also became nearly impossible to displace. Yet, over the past several years I have had many informal discussions with a variety of scientists who work on terrestrial fossils, all expressing grave doubts about a terrestrial extinction – in many instances, not so much that it did not occur, but that the terrestrial record, as presently constituted, is too poor to provide a definitive answer.

Here, I believe, the authors present a data set that addresses these matters squarely. Pollen and spores, unlike any other terrestrial fossils, plant or animal, have a high preservation potential, are extremely widely dispersed, and most often are drawn from broad areas of the terrestrial landscape. They require a lot of specialist training to interpret, which is why such data have not been synthesized in much of the geological record (with notable exceptions). The authors, furthermore, make their case without hyperbole and without engaging in the nastiness that seems to be part of much of the Permian-Triassic boundary literature.

It is my personal experience with the terrestrial plant macrofossil record that it has significant biases. These biases, expressed in several papers (I would point to papers in the plant taphonomy literature by Burnham, Gastaldo, Pfefferkorn, Falcon-Lang, and others) and in my own writing (the authors cite a recent paper by Looy et al.), indicate clearly that there are often enormous discontinuities/gaps in the distribution of taxa, often at the genus and species levels of resolution, amounting from millions to 10s of millions of years long. **[emphasis added]** There are now many reports in the literature not only of so-called “Lazarus” taxa (reappearances long after the organisms had been thought to be extinct), but also well before they were thought to have originated (what we’ve dubbed “Methuselah taxa”). These kinds of patterns do not mean the terrestrial record cannot be used to address interesting problems in biology. What they indicate is that the terrestrial record must be used judiciously and in ways commensurate with its strengths and weaknesses. – I believe this paper does just that.

→ **We are thankful for the reviewer’s endorsement. We also appreciate the mention of plant taphonomy literature, which led us to extend the discussion on this topic, including references to works of the mentioned authors (lines). Following also a suggestion by Reviewer #1, the cited reference by Looy et al. was replaced by DiMichele and Gastaldo 2008³⁵.**

Detailed comments are keyed to line number. They are primarily editorial. I find little to disagree with in the science presented here. This is about as “tight” a study as one might ever hope for.

Line 10: delete "following"

→ **Deleted as suggested.**

Lines 15-16: I suggest rewording "the taxonomic...plants)" to say "the taxonomic diversities of gymnospermous macrofossils and of the pollen they produced"

→ **This was rephrased as: “The taxonomic diversities of gymnosperm macrofossils and of the pollen produced by this group [...].” (lines 15-16).**

Line 17: It is unclear from the wording here if gymnosperm macrofossils are undersampled, or if all

gymnosperm fossils, macro and micro, are undersampled. This should be clarified because a distinction has been made between these earlier in the paragraph.

→ **We changed “gymnosperms” to “gymnosperm macrofossils” (line 17).**

Line 33: Change "On the southern hemisphere" to "In the Southern Hemisphere"

→ **Changed as suggested (line 34).**

Line 37: Change "account for" to "be accounted for by"

→ **Changed as suggested (line 38).**

Line 43: Change: "unused, as in most cases" to "unused and, as in most cases,"

→ **Changed to “unused, for in most cases,” (line 44).**

Line 46: I suggest changing "has been attempted" to "is presented here". "Has been attempted" is a very equivocal statement, and carries with it an implication of having tried and failed. If some equivocation is necessary, then say that the study "is presented here, based on presently available data".

→ **Changed to “is presented here” (line 47).**

Line 56: Is this the first use of the acronym "PTB"? Perhaps it should be first used parenthetically in line 53, following the words "Permian-Triassic boundary".

→ **This was in fact the only occurrence of this abbreviation. We exchanged it with the long version: “Permian–Triassic boundary” (lines 57-58).**

Line 64: Change "which contrasts the" to "which contrasts with the"

→ **Changed as suggested (line 72).**

Line 69: It would be better to say "approach to partitioning" instead of "approach in partitioning".

→ **Changed as suggested (line 77).**

Lines 73-74. I might add here that sporomorph diversity is higher than macrofossil diversity because of the transport potential of spores and pollen, by both wind and water, which draws in elements from a wider diversity of habitats than macrofossils, and also has the potential to sample the regional flora. These are very important points.

→ **The sentence was extended: “The diversity of sporomorph species and genera is generally higher compared to macrofossil diversity (Fig. 1), which is expected due to the generally higher preservation potential of sporomorphs, their transportability by wind and water even from distant habitats to favourable depositional settings, their sheer abundance and the possibility of a single plant species to produce multiple spore or pollen taxa.” (lines 80-84).**

Line 87: change "unknown, however," to "unknown. However,"

→ **Changed as suggested (line 100).**

Lines 88-90. rewrite

Replace the semi-colon after categories with a colon.

Delete "are" after "pollen" and replace it with a comma

Delete "are" after "iso-/microspores" and replace with a comma

Delete "are" after "megaspores" and replace with a comma

→ **Changed as suggested (lines 101-102).**

Line 96: This is just a suggestion, here and elsewhere in the manuscript. Because of the complexities of botanical terminology, and although "everybody knows what you mean", it is technically incorrect to call non-seed plants "spore plants" (seed plants still produce spores, we just have specialized names for them). It would be better to refer to the "spore plants" as "pteridophytes" or "pteridophytes and bryophytes" (or even "pteridophytes, bryophytes, and fungi")

→ **A clarification was added at the first instance: "spore plant (pteridophyte, lycophyte, sphenophyte and bryophyte)" (line 109). Fungi are excluded from the analysis.**

Line 101: I suggest replacing "already in" with "by"

→ **Changed as suggested (line 115).**

Lines 110-113. It actually is not clear to me from the titles of the Excel data sheets which one of these is Table 4, but it would seem to be the one with all the correlations in it.

That said, in lines 110 & 111 it is stated that there is a strong correlation between "the number of entries per stage" and "genus diversity". OK, that is very clearly stated, and understood without having to look at the Table for myself. In lines 112 & 113, however, it is stated that there is a less strong correlation "of the data distribution" with "macrofossil species diversity" and with "sporomorph genus diversity". I cannot follow/understand this statement, or interpret it from, what I presume to be, Table 4. What is "the data distribution"? Number of samples per time bin? This is crucial and must be made more clear; if it is sample number per time bin, then I suggest using the same language as above: "the number of entries per stage". If it is something else, then clarification is needed.

→ **The file names of the supplementary tables were altered by the submission system, but the reviewer appears to have identified the correct one. Regarding "data distribution", we added a clarification in the main text that this indeed refers to the number of entries per stage (line 127), and also a more detailed explanation in the Methods section (lines 298-300).**

Line 120: Change "least" to "lowest"

→ **The sentence was rephrased entirely (lines 135-137).**

Line 121: Change "are not" to "have not been"

→ **Changed as suggested (line 138).**

Lines 141-142: I suggest putting a comma after "signal", changing "does" to "do", and putting "mass extinction" in quotation marks.

→ **Changed as suggested (lines 166,168).**

Line 194: Change "constraint dating" to "constrained date"

→ **Changed as suggested (line 239).**

Line 202: "total diversity", as it is used in this sentence, is often referred to as "range-through" diversity, because ranges of taxa are extended through intervals where their actual occurrence has not been documented. I do not think "total diversity" is a term that is generally or systematically used interchangeably with range-through diversity.

→ **The term "total diversity" in this sense has been used e.g. by Foote³¹⁷, Cooper⁶⁹, Rabosky and Sorhannus³¹⁸. The term "range-through diversity" refers to the "range-through method"⁶⁸ (or "range method"⁶⁷) for calculating diversity, as opposed to the "occurrence method"⁶⁷ (producing sampled-in-bin diversity). They are conceptually interchangeable, but we prefer "total diversity" as the more intuitive term and to avoid confusion with the number of range-through taxa. We extended the explanation in the methods section to emphasize the connection between the total diversity index and the range-through method (lines 258-261,268).**

End of comments

William A. DiMichele

Department of Paleobiology

NMNH Smithsonian Institution

Washington, DC 20560

dimichel@si.edu

Reviewer #3 (Remarks to the Author):

This paper revisits changes in plant diversity across the PT mass extinction, proposing provocatively that the extinction had almost no substantial impact on plants. This is a paper of potential broad interest, and with important implications. Unfortunately, the validity and significance of the work cannot be assessed because the underlying data has not been made available. Moreover, the taxon-counting approach used does not take into account the severe biases that exist in the fossil record and the results might be flawed as a result, and there are also problems with the statistical analyses.

My number one concern is that the dataset underlying the study, which is arguably the most significant part of the paper, is not being made available to other researchers. The methods section states that the database is available from the corresponding author "upon reasonable request". This is unacceptable, as it makes the research unrepeatable - who decides what a "reasonable" request is, and what happens if the corresponding author leaves the research field or is no longer contactable. Any acceptance of this manuscript should be contingent on the raw data being made immediately available to readers, in line with current standards within the research field, as well as my reading of journal policy:

"Authors are required to make materials, data, code, and associated protocols promptly available to readers without undue qualifications."

→ **We received confirmation from the editor that the earlier data availability statement was acceptable under the journal's policy. However, we do understand the reviewer's concerns and the demand for easily accessible data. Therefore, the raw data used in this study and code**

to reproduce the results are now included in the supplementary information (Supplementary Tables 2,3 and Supplementary Code).

I also have a variety of other comments, detailed below:

- The methods consist of raw taxon counts, but there is a vast literature stretching back >40 years that has shown beyond all reasonable doubt that the fossil record is highly biased and that raw, observed counts of taxa are highly likely to be misleading. An array of different statistical approaches - rarefaction, shareholder-quorum subsampling, diversity residuals, TRiPS, phylogenetic diversity estimates etc. - have been developed to deal with this problem, and try to accurately estimate diversity change. This manuscript does not use any of these methods, but there is no discussion of why not, and why the plant fossil record apparently can be read at face value. The authors should either engage with these methods, or provide explicit justification for their approach.

→ **We did not make the claim nor wished to imply that the plant fossil record can be taken at face value, but it often has been in the past. The basic idea of the current work was in fact to see to what degree the plant fossil record can be taken at face value specifically with respect to the Permian–Triassic transition, and to identify specific problems. The reviewer is correct in saying that there are various methods for dealing with such problems. However, the interpretation of the results of these methods is not without challenges itself. We have included shareholder quorum subsampling in the revised manuscript (Fig. 3; lines 64–66,96–97,282–296,561–564; Supplementary Code), as a method for estimating taxonomic richness that is currently widely used in similar works. The results were in line with our earlier analysis, but showed a very large confidence interval. This makes it impossible to draw meaningful conclusions about the actual diversity, but serves to illustrate the difficulties that the properties of the data itself present on top of the requirement to choose an appropriate statistical method. We consider the problem of accurately estimating diversity to be sufficiently distinct and broad to warrant excluding it largely from the present paper. That said, we have provided the raw data in the supplementary information (Supplementary Tables 2,3) so that others can try their method of choice on the dataset as well.**

- The authors repeatedly refer to certain stages being 'quantitatively underestimated' or having poor 'coverage', yet no information is provided on how this is defined. 'Coverage' is a term with a precise meaning in diversity analyses (see papers by John Alroy on SQS), but I do not think the authors are using the term with the same meaning.

→ **We believe the term can be used more broadly, but to avoid confusion, we added an explanation of the term “sampling coverage” in Good’s sense⁷³ (which also John Alroy’s papers^{70,71} refer to) in the methods section (lines 284–293) and ensure that it is used consistently in the text (see lines 95,137). Furthermore, Good’s u has been included in the results as a metric for estimating sampling coverage (Fig. 3, Supplementary Table 5).**

- The authors do not explain how they calculate extinction and origination rates. They also do not really explain why greatly elevated macrofloral extinction rates in the Induan do not correspond to a mass extinction.

→ **The calculation of extinction, origination and turnover rates was mentioned in the methods section, but the explanation was probably too brief. This has been extended for more clarity (lines).**

- Genus diversity is compared to 'number of entries' per stage to assess sampling biases, but it is completely unclear what 'number of entries' means, in part because the data is not available.

→ **The raw data is now available as part of the supplementary information (Supplementary Tables 2,3). The meaning of entries is now furthermore explained in the methods section (lines 298-300).**

- The normal threshold for statistical significance is $\alpha = 0.05$. However, the authors refer to $p = 0.058$ as "significant", and still apparently consider $p = 0.14$ as significant, albeit "less significant", whereas this would definitely be considered non-significant by other researchers. Clarification of the stats is required.

→ **The text was changed to clearly distinguish statistically significant and insignificant correlations and the chosen threshold (0.05) is declared in the methods section (lines 304-305).**

- Comparisons of time series should use approaches that account for serial correlation (e.g. generalised least squares), which can inflate correlation coefficients. The authors should apply such approaches for their comparisons of "entries per stage" and diversity.

→ **The entries per stage do not actually represent a time series, since they were not generated in the order of the stratigraphic succession. By extension, the *sampled* diversity, which is derived from the data, is not a "pure" time series (it is conceptually distinct from the "true" *standing* diversity, which would be a time series). This and the low amount of data points makes the use of generalised least squares problematic. Autocorrelation is certainly an issue to be taken into account, but in this case, we believe that spatial autocorrelation would be more important due to the clustering of sampling localities. This however goes beyond the scope of the current work.**

- "Lazarus" taxa are discussed but their abundance does not appear to be shown in any of the plots.

→ **They are now presented in Fig. 1.**

- Changes in diversity are often referred to as 'not catastrophic', 'slight' etc., but with no quantification of what these terms mean. Percentage diversity changes should be provided, and terms should be qualified. What do you mean by 'catastrophic'? What is your definition of a mass extinction, and why don't the diversity changes observed here comprise one? Explicit, quantitative discussion of this is required.

→ **Unfortunately, terms such as "mass extinction" are usually used without a strict definition. A quantifiable definition exists, but is not applicable in this case. This issue is now also discussed in the main text (lines 159-162). We furthermore added the percentage of genus extinctions for macrofossil (19 %) and sporomorph genera (17 %) to the discussion (lines 166-167).**

- Figures are not sufficiently often referred to in the text, and so it can be difficult to tell which figure should be examined (e.g. no figure references at all in lines 85-93).

→ **We added multiple references to the figures (and supplementary files) throughout the text.**

Reviewer #4 (Remarks to the Author):

This is a very important paper that uses large and new data sets to analyze what happened to land plants at the Permian-Triassic transition. These results are highly significant because they show that plants reacted differently from land animals and marine invertebrates. These results correct some earlier studies that were based on limited data sets or the use of unsuitable taxonomic levels in plants.

Suggested title: No mass extinction for plants at the Permian-Triassic transition

→ **We appreciate the suggestion and did reconsider the title. We settled on: “No mass extinction: diversity patterns of land plants across the Permian–Triassic boundary”.**

The last sentence in the abstract beginning with “This means that none ...” is an overstatement and should be eliminated. This is not a conclusion that can be drawn from this paper.

→ **The sentence has been deleted.**

Lines 37-38 should be changed to: ..., but it could also be accounted for by a severe taphonomic bias.

→ **Changed as suggested (lines 38-39).**

Lines 66-68: the paper by Cascales-Miñana & Cleal uses families and singletons and their conclusions are wrong. There is no mass extinction at the Carboniferous-Permian boundary and what they say about the P-Tr boundary is wrong, too. The most stable taxonomic level for plant macrofossils is the genus. The attribution to families depends on school and is often still artificial. There are also genera not yet attributable to any family.

→ **We agree that genus level is the most appropriate.**

Line 171: From the studies of Conrad Labandeira and his colleagues we know that insects and plants had mutualistic relationships back in the Pennsylvanian. Are we certain that this was not also the case for at least some vertebrates? The phrase “Such mutualistic dependencies .. “ is a strong statement that might not be correct and has no function here.

→ **The reviewer makes a valid point. While the existence of plant-insect mutualism, specifically pollination, in the Permian is not unequivocal and would apparently have been independent from younger instances, the previous statement was indeed too strong. The previous statement was replaced by the following, which takes into account the studies mentioned by the reviewer: “Such mutualistic dependencies may have been in effect between certain insects and some late Palaeozoic pteridospermatophytes and conifers⁵⁷⁻⁶⁰. The insects involved died out in the end-Permian mass extinction, and modern forms of mutualism developed during the Mesozoic⁵⁷⁻⁶⁰.” (lines 210-213).**

In Figures 1 and 2 the same scale should be used in each graph wherever possible. It is clear that it cannot be done in all cases.

→ **Figure 2 now uses the same scale for all graphs. In Fig. 1, the displayed indices have very different dimensions, so that the use of equal scales would have required a tradeoff between visibility and space. We opted to prioritize visibility, compactness and comparability of trends rather than absolute values in Fig. 1, so that the graphs have different scales but similar graphical dimensions.**

Line 370: period and space missing before g

→ **Corrected (line 586).**

This manuscript demonstrates that several “things” happened to plants at the P-Tr transition but it was not a mass extinction. That is the important conclusion. The “things” that happened to plants were very well summarized in the publication by McElwain & Punyasena (2007). Perhaps the other aspect of the “not mass extinction” should be discussed more to give a more rounded account.

→ **The discussion was extended as suggested (lines 189-197) and several references were added (refs. 42-50).**

References

5. Cascales-Miñana, B. & Cleal, C. J. The plant fossil record reflects just two great extinction events. *Terra Nova* **26**, 195–200 (2014).
7. Rees, P. M. Land-plant diversity and the end-Permian mass extinction. *Geology* **30**, 827–830 (2002).
8. Lucas, S. G. Permian tetrapod extinction events. *Earth-Sci. Rev.* **170**, 31–60 (2017).
9. Neveling, J. *et al.* A Review of Stratigraphic, Geochemical, and Paleontologic Data of the Terrestrial End-Permian Record in the Karoo Basin, South Africa. in *Origin and Evolution of the Cape Mountains and Karoo Basin* 151–157 (Springer, Cham, 2016). doi:10.1007/978-3-319-40859-0_15
10. Gastaldo, R. A. & Neveling, J. Comment on: “Anatomy of a mass extinction: Sedimentological and taphonomic evidence for drought-induced die-offs at the Permo–Triassic boundary in the main Karoo Basin, South Africa” by R.M.H. Smith and J. Botha-Brink, *Palaeogeography, Palaeoclimatology, Palaeoecology* 396:99-118. *Palaeogeogr. Palaeoclimatol. Palaeoecol.* **447**, 88–91 (2016).
11. Gastaldo, R. A. & Neveling, J. The terrestrial Permian–Triassic boundary event is a nonevent: REPLY. *Geology* **40**, e257 (2012).
12. Gastaldo, R. A., Neveling, J., Clark, C. K. & Newbury, S. S. The terrestrial Permian–Triassic boundary event bed is a nonevent. *Geology* **37**, 199–202 (2009).
13. Gastaldo, R. A. *et al.* Is the vertebrate-defined Permian-Triassic boundary in the Karoo Basin, South Africa, the terrestrial expression of the end-Permian marine event? *Geology* **43**, 939–942 (2015).
35. DiMichele, W. A. & Gastaldo, R. A. Plant paleoecology in deep time. *Ann. Mo. Bot. Gard.* **95**, 144–198 (2008).
42. Looy, C. V., Collinson, M. E., van Konijnenburg-van Cittert, J. H. A., Visscher, H. & Brain, A. P. R. The ultrastructure and botanical affinity of end-Permian spore tetrads. *Int. J. Plant Sci.* **166**, 875–887 (2005).
43. Eshet, Y., Rampino, M. R. & Visscher, H. Fungal event and palynological record of ecological crisis and recovery across the Permian-Triassic boundary. *Geology* **23**, 967–970 (1995).

44. Visscher, H. *et al.* The terminal Paleozoic fungal event: evidence of terrestrial ecosystem destabilization and collapse. *Proc. Natl. Acad. Sci.* **93**, 2155–2158 (1996).
45. Visscher, H., Sephton, M. A. & Looy, C. V. Fungal virulence at the time of the end-Permian biosphere crisis? *Geology* **39**, 883–886 (2011).
46. Elsik, W. C. *Reduviasporonites* Wilson 1962: Synonymy of the fungal organism involved in the late Permian crisis. *Palynology* **23**, 37–41 (1999).
47. Rampino, M. R. & Eshet, Y. The fungal and acritarch events as time markers for the latest Permian mass extinction: An update. *Geosci. Front.* **9**, 147–154 (2018).
48. Afonin, S. A., Barinova, S. S. & Krassilov, V. A. A bloom of *Tympanicysta* Balme (green algae of zygnematalean affinities) at the Permian–Triassic boundary. *Geodiversitas* **23**, 481–487 (2001).
49. Foster, C. B., Stephenson, M. H., Marshall, C., Logan, G. A. & Greenwood, P. F. A Revision of *Reduviasporonites* Wilson 1962: Description, Illustration, Comparison and Biological Affinities. *Palynology* **26**, 35–58 (2002).
50. Sephton, M. A., Visscher, H., Looy, C. V., Verchovsky, A. B. & Watson, J. S. Chemical constitution of a Permian–Triassic disaster species. *Geology* **37**, 875–878 (2009).
57. Labandeira, C. C. How old is the flower and the fly? *Science* **280**, 57–59 (1998).
58. Labandeira, C. C. Early history of arthropod and vascular plant associations. *Annu Rev Earth Planet Sci* **26**, 329–377 (1998).
59. Labandeira, C. C. The Paleobiology of Pollination and its Precursors. *Paleontol. Soc. Pap.* **6**, 233–270 (2000).
60. Labandeira, C. The origin of herbivory on land: Initial patterns of plant tissue consumption by arthropods. *Insect Sci.* **14**, 259–275 (2007).
67. Cheetham, A. H. & Deboo, P. B. A numerical index for biostratigraphic zonation in the mid-Tertiary of the Eastern Gulf. *Gulf Coast Assoc. Geol. Soc. Trans.* **13**, 139–147 (1963).
68. Boltovskoy, D. The range-through method and first-last appearance data in paleontological surveys. *J. Paleontol.* **62**, 157–159 (1988).
69. Cooper, R. A. Measures of diversity. in *The Great Ordovician Biodiversification Event* (eds. Webby, B. D., Paris, F., Droser, M. L. & Percival, I. G.) 52–57 (Columbia University Press, 2004).
70. Alroy, J. Accurate and precise estimates of origination and extinction rates. *Paleobiology* **40**, 374–397 (2014).
71. Alroy, J. Fair sampling of taxonomic richness and unbiased estimation of origination and extinction rates. *Quant. Methods Paleobiology Paleontol. Soc. Pap.* **16**, 55–80 (2010).
73. Good, I. J. The population frequencies of species and the estimation of population parameters. *Biometrika* **40**, 237–264 (1953).
209. Silvestro, D., Cascales-Miñana, B., Bacon, C. D. & Antonelli, A. Revisiting the origin and diversification of vascular plants through a comprehensive Bayesian analysis of the fossil record. *New Phytol.* **207**, 425–436 (2015).
317. Foote, M. Origination and extinction components of taxonomic diversity: general problems. *Paleobiology* **26**, 74–102 (2000).
318. Rabosky, D. L. & Sorhannus, U. Diversity dynamics of marine planktonic diatoms across the Cenozoic. *Nature* **457**, 183–186 (2009).

Reviewers' Comments:

Reviewer #1:

Remarks to the Author:

Novak et al. have revised the current manuscript according to the recommendations made by the reviewers, and have added necessary text both in the body of the work and the figure captions to clarify those aspects of the previous contribution found to be ambiguous. Additional citations also have been added to the current version, enhancing the coverage of pertinent literature about the end-Permian and early Triassic. Additions to the Method section, both in the body and supplemental materials (raw data and computer code), makes the work more transparent and one that can be tested by others in the future, if so desired, using other methodologies and analytical techniques. This is a very positive aspect of the current version. The paper continues to be concise and focused, providing the first attempt at utilizing a comprehensive data set to assess what has been considered, by many paleontologists, a trend in the terrestrial plant-fossil record that mirrors the marine mass extinction. The results of the current analysis, in contrast to the prevailing wisdom, demonstrates that land plants did not suffer widespread extinction, with a suite of subsequent originations to fill empty ecospace. The paper is well developed and well executed, and is an exciting analysis based on a very large, global data set covering both the megafloreal and microfloreal records. I find the revised version to warrant publication.

Reviewer #3:

Remarks to the Author:

I thank the reviewers for engaging constructively with my review, and most importantly for making the underlying data available. I am happy to recommend that this paper progress now to publication - it is an important and provocative study that is sure to be of broad interest and generate substantial debate.

REVIEWERS' COMMENTS:

Reviewer #1 (Remarks to the Author):

Novak et al. have revised the current manuscript according to the recommendations made by the reviewers, and have added necessary text both in the body of the work and the figure captions to clarify those aspects of the previous contribution found to be ambiguous. Additional citations also have been added to the current version, enhancing the coverage of pertinent literature about the end-Permian and early Triassic. Additions to the Method section, both in the body and supplemental materials (raw data and computer code), makes the work more transparent and one that can be tested by others in the future, if so desired, using other methodologies and analytical techniques. This is a very positive aspect of the current version. The paper continues to be concise and focused, providing the first attempt at utilizing a comprehensive data set to assess what has been considered, by many paleontologists, a trend in the terrestrial plant-fossil record that mirrors the marine mass extinction. The results of the current analysis, in contrast to the prevailing wisdom, demonstrates that land plants did not suffer widespread extinction, with a suite of subsequent originations to fill empty ecospace. The paper is well developed and well executed, and is an exciting analysis based on a very large, global data set covering both the megafloreal and microfloral records. I find the revised version to warrant publication.

Reviewer #3 (Remarks to the Author):

I thank the reviewers for engaging constructively with my review, and most importantly for making the underlying data available. I am happy to recommend that this paper progress now to publication - it is an important and provocative study that is sure to be of broad interest and generate substantial debate.

→ **We are grateful for the reviewers' approval.**